# Ulvan-Na, an Ulvan Subjected to Na^+^ Cation Exchange, Improves Intestinal Barrier Function in Age-Related Leaky Gut

**DOI:** 10.3390/md23100390

**Published:** 2025-09-30

**Authors:** Yuka Maejima, Yuki Morioka, Yusei Sato, Masanori Hiraoka, Ayumu Onda, Takushi Namba

**Affiliations:** 1Department of Marine Resource Science, Faculty of Agriculture and Marine Science, Kochi University, Oko-cho Kohashu Nankoku-shi, Kochi 783-8502, Japan; b24m6h39@s.kochi-u.ac.jp (Y.M.); b24m6h44@s.kochi-u.ac.jp (Y.M.); b25m6h39@s.kochi-u.ac.jp (Y.S.); 2Usa Marine Biological Institute, Kochi University, 194 Inoshiri, Usa, Kochi 781-1164, Japan; mhiraoka@kochi-u.ac.jp; 3Graduate School of Kuroshio Science, Kochi University, 2-5-1 Akebono-cho, Kochi 780-8520, Japan; 4Research and Education Faculty, Multidisciplinary Science Cluster, Interdisciplinary Science Unit, Kochi University, Oko-cho Kohashu Nankoku-shi, Kochi 783-8505, Japan; aonda@kochi-u.ac.jp; 5Ulva. Co., Ltd., 194 Inoshiri, Usa, Kochi 781-1164, Japan

**Keywords:** *Ulva meridionalis*, ulvan, polysaccharide, leaky gut, tight junction, microbiome, β-catenin, aging

## Abstract

The global increase in life expectancy underscores the need to promote healthy aging, particularly by addressing age-related leaky gut syndrome, which contributes to systemic inflammation and chronic disease. This study focused on the sustainable production and functional development of *Ulva meridionalis*, a fast-growing seaweed, to improve gut health and mitigate the effects of aging. Using land-based aquaculture, a scalable cultivation system for *U. meridionalis* was established, and its polysaccharide, ulvan, was extracted. Ion exchange treatment enhanced the functionality of ulvan to produce ulvan-Na, which contains high levels of Na^+^ and conveys superior anti-aging properties. Ulvan-Na restored intestinal barrier integrity in aged mice by reducing serum LPS levels and increasing claudin-1 expression. Ulvan-Na modulated the gut microbiota, increasing beneficial bacteria such as *Clostridiales vadin BB60* and suppressing inflammatory bacteria such as *Turicibacter*. The mechanism was clarified whereby ulvan-Na activates β-catenin to enhance claudin-1 expression. These findings highlight ulvan-Na as a bioactive compound that ameliorates age-related intestinal dysfunction while demonstrating the feasibility of sustainable *U. meridionalis* production for functional food innovation and environmental conservation.

## 1. Introduction

The global average life expectancy is 73 years and is projected to increase, according to the World Health Organization (WHO). However, considerable disparities exist among countries, with higher life expectancies observed in regions such as Japan and Western Europe compared to lower values in parts of sub-Saharan Africa. In addition, healthy life expectancy is approximately 10 years shorter than the average life expectancy worldwide, underscoring the importance of strategies to extend years lived in good health. Addressing these disparities aligns with the United Nations Sustainable Development Goals of the 2030 Agenda, particularly the goal of ensuring healthy lives and promoting well-being for all at all ages [1,2]. In countries with declining birth rates and aging populations, increasing healthy life expectancy is critical for reducing healthcare costs and maintaining economic productivity [3].

Aging is a complex biological process characterized by a decline in physiological function and an increased risk of various age-related diseases [4]. In recent years, there has been a growing effort to treat aging as a disease, exemplified by the development of senolytics [5]. Among the factors receiving attention are the relationships between intestinal abnormalities, known as “leaky gut,” and age-related diseases. “Leaky gut” refers to a condition in which intestinal barrier function is compromised due to factors such as aging, stress, and dysbiosis of the gut microbiota, thereby allowing large molecules, toxins, and pathogens to cross the intestinal wall into the bloodstream [6]. This can lead to systemic inflammation and has been implicated in the onset and exacerbation of autoimmune diseases, metabolic syndrome, and neurodegenerative diseases [7].

The gut microbiota and gut barrier play critical roles in maintaining a healthy gut environment. A balanced gut microbiota is essential for proper digestion, nutrient absorption, and immune function [8]. However, aging leads to significant changes in the composition and diversity of the gut microbiota, resulting in dysbiosis [9]. This, in turn, exacerbates the deterioration of intestinal barrier function and leaky gut syndrome [6]. Tight junctions are key components of intestinal barrier function, with claudin family proteins expressed at the apical region being of particular importance [10,11]. In the aged intestine, the expression of barrier-related claudins, such as claudin-1 and claudin-4, decreases, whereas the expression of channel-related claudins increases [12,13]. The loss of barrier-related claudin expression allows various molecules to enter the bloodstream through the intercellular spaces of the intestinal epithelium. It has been suggested that inflammatory responses are induced in the aged intestine, and inflammation suppresses the expression of barrier-related claudins [14]. Certain probiotics and prebiotics have been shown to improve the diversity of the gut microbiota and strengthen gut barrier function [15]. In addition, plant-derived bioactive compounds such as polyphenols and polysaccharides with anti-inflammatory and antioxidant properties have been shown to improve the gut environment [16,17]. However, although changes in the intestinal microbiota induced by these compounds have been studied, the molecular mechanisms underlying the effects of these compounds on intestinal cells are largely unknown. Therefore, the discovery and development of dietary health-promoting materials that ameliorate age-related leaky gut are highly desirable.

Most natural health food ingredients are plant-derived, yet climate change and overharvesting threaten about 45% of known angiosperms and 75% of unrecorded vascular plant species [18,19]. To protect biodiversity, health-promoting materials should be developed from sustainable, environmentally friendly, and non-region-specific sources.

Seaweed grows by absorbing and fixing carbon dioxide via photosynthesis, is more efficiently fertilized than land plants, and is a sustainable material that can be produced without harming the environment [20]. Although Japan has a long history of seaweed aquaculture and technological development, traditional ocean-based methods are declining due to environmental changes such as global warming. Land-based aquaculture, exemplified by the Kochi method [21], offers a solution. For instance, cultivation of the green alga *Ulva prolifera* produces over 20 tons (dry weight) annually at multiple sites. Thus, land-based aquaculture is gaining attention both as an alternative to unstable ocean farming and as a labor-saving, sustainable method of seaweed production.

*Ulva meridionalis*, a green alga of the genus *Ulva*, is a very fast-growing seaweed that can quadruple its weight in a day [22]. It fixes a carbon dioxide content equivalent to its weight gain in polysaccharides; thus, its mass cultivation and use as a material can promote carbon cycling in the environment and help reduce carbon dioxide levels [23]. In addition, since its optimal growth temperature is approximately 30 °C, it can be cultivated as an adaptation to ongoing global warming [22]. To increase the production of fast-growing *U. meridionalis* without releasing it into the ocean or causing environmental impacts, it is important to establish land-based cultivation methods. In addition, the use of polysaccharides, whose main component is carbon derived from carbon dioxide, is also important. The polysaccharide synthesized by *Ulva* spp. is called ulvan, a sulfated polysaccharide composed mainly of rhamnose (Rha), glucuronic acid (GlcA), iduronic acid (IdoA), and sulfate groups [24]. Previous reports have suggested that ulvan has immunostimulatory effects and influences the gut microbiota, but unlike fucoidan, a polysaccharide derived from brown algae, it has not been commercialized [25]. *U. meridionalis* also produces ulvan, but its physiological activity remains unknown. Furthermore, the physiological activities of ulvan modified through ion exchange remain unexplored.

In this study, *U. meridionalis* was focused on as a sustainable material, and land-based open-air aquaculture methods were investigated to achieve its stable production. Subsequently, the development of a health-promoting material derived from *U. meridionalis* polysaccharides, specifically ion-exchanged ulvan, was pursued to improve age-related leaky gut symptoms.

## 2. Results

### 2.1. Production of U. meridionalis Through Land-Based Cultivation

*U. meridionalis*, similar in morphology to *U. prolifera* in terms of its thin and branched structure, was identified as a novel species in 2011 on the basis of microscopic cell morphology and DNA marker comparisons [26]. Since the mass of *U. meridionalis* can increase up to fourfold in a single day under optimal conditions, the establishment of a stable mass aquaculture system and methods for its utilization as biomass are crucial. Therefore, land-based cultivation methods for *U. meridionalis* were investigated. Land-based cultivation techniques for the green alga *U. prolifera* have previously been established and commercialized. With these methods, land-based cultivation of *U. meridionalis* was investigated. The system was configured to automatically drain excess seawater to maintain a closed cultivation environment (Appendix A). Appendix A shows the growth curves of *U. meridionalis* in the 1-ton tanks over two periods. In period 1, the seawater temperature was 20.5 ± 1.4 °C, and the average daily growth rate was 2.2 ± 0.3-fold. In period 2, the temperature was 21.1 ± 2.1 °C, and the average daily growth rate was 2.6 ± 0.3-fold, indicating efficient cultivation. An image of the harvested *U. meridionalis* is shown in Appendix A. These results demonstrate that mass cultivation of *U. meridionalis* using the germling cluster method is feasible.

### 2.2. Purification of Ion-Exchanged Ulvan from U. meridionalis

Previous reports have indicated that *U. meridionalis* fixes CO_2_-derived carbon in polysaccharides [23]. The effective utilization of these polysaccharides would, therefore, contribute to carbon dioxide mitigation efforts. The harvested *U. meridionalis* (Appendix A) was washed with tap water, dehydrated, dried, pulverized, and subjected to hot-water extraction to obtain water-soluble polysaccharides. It has previously been reported that the monosaccharide composition of ulvan from the *U. meridionalis* strain used in this study consisted mainly of Rha, glucose (Glc), xylose (Xyl), mannose (Man), and galactose (Gal) and remained consistent even under varying cultivation conditions [23]. While various physiological activities of ulvan have been described at a basic research level, ulvan has not yet been commercialized as a health food ingredient, and the functional properties of ulvan derived from *U. meridionalis* remain unexplored. Considering the potential anti-aging effects of polysaccharides from green algae, ulvan was first investigated for such properties, but no significant effects were observed across multiple indicators (Figure 1). Previous studies have shown that altering the cations present in seaweed-derived polysaccharides can alter their physical properties. Therefore, it was hypothesized that modifying the cationic composition of ulvan could improve its functionality. As shown in Table 1, the primary cations found in ulvan are Na^+^, K^+^, Ca^2+^, and Mg^2+^, with Mg^2+^ being the most abundant. Using ion-exchange chromatography, the cationic composition was adjusted to increase the abundance of monovalent cations (Na^+^, H^+^, K^+^), creating ion-exchanged ulvan variants. It was then evaluated whether these modifications altered the anti-aging effects compared with those of the wildtype ulvan.

### 2.3. Investigation of the Anti-Aging Effects of Ion-Exchanged Ulvan

The anti-aging effects of ulvan and ion-exchanged ulvan were evaluated using replicative, senescent NB1RGB human diploid fibroblasts [27]. First, the impact of ulvan on the activity of senescence-associated β-galactosidase (SA-β-gal), a marker of cellular aging, was examined. As shown in Figure 1A, treatment with ulvan did not affect the number of SA-β-gal-expressing cells. However, treatment with ulvan-H or ulvan-Na significantly reduced the number of SA-β-gal-expressing cells.

Next, the effects of ulvan on ATP production and collagen synthesis in senescent cells were investigated. While ulvan treatment resulted in no significant changes, ion-exchanged ulvan treatment resulted in a significant increase in both ATP production and collagen synthesis compared with those in control (distilled water) and ulvan-treated cells (Figure 1B,C). The effects of ulvan-Na samples with different sodium contents were also evaluated, showing that the anti-aging effects did not differ significantly between the 41 mol% and 93 mol% Na^+^ exchange levels. These results suggest that ion-exchanged ulvan has anti-aging properties.

Previous studies indicated that upregulation of the expression of SOD2, an enzyme that eliminates mitochondrial reactive oxygen species, is critical for its anti-aging effects [27]. Whether ion-exchanged ulvan could induce SOD2 expression was investigated. The results revealed that while ulvan did not affect SOD2 expression, ion-exchanged ulvan significantly induced SOD2 expression (Figure 1D). Regarding sodium ions, 30 mol% exchange was found to be less effective than 41 mol% exchange in inducing SOD2 expression. Similarly, regarding hydrogen ions, 35 mol% exchange was less effective than 50 mol% exchange. For potassium ions, a 69 mol% exchange was more effective than a 59 mol% exchange. These results suggest that the degree of ion exchange correlates with the degree of SOD2 mRNA expression induction, with Na^+^ ion exchange being particularly effective even at lower exchange levels. Since the strongest functionality was observed at Na^+^ exchange levels of 41 mol%, ulvan-Na with 41 mol% Na^+^ was selected for subsequent experiments. Additionally, ulvan and ulvan-Na exhibit distinct water absorption capacities, with ulvan-Na demonstrating superior water absorbency (Appendix A). This difference in material properties may influence their functionality.

### 2.4. Verification of a Mouse Model of Aging-Dependent Leaky Gut

Because seaweed-derived polysaccharides are indigestible and unlikely to be systemically absorbed, the gut was considered the primary site of their direct effects. Aging impairs various organ functions, and one consequence of intestinal aging is increased intestinal permeability, or “leaky gut.” While leaky gut has been studied using chemical induction (e.g., with DSS or acetic acid), genetically modified models (e.g., claudin-deficient models), and diet-induced models (e.g., with high-fat or high-sugar diets), research on leaky gut resulting solely from natural aging is limited [28,29,30]. Whether naturally aged mice exhibit changes in intestinal integrity was first investigated. A hallmark of leaky gut is the translocation of bacterial lipopolysaccharide (LPS) into the bloodstream due to impaired intestinal barrier function, as well as increased permeability to orally administered fluorescent FITC-dextran [30]. In aged mice, oral 4 kDa FITC-dextran did not increase serum IL-6 levels beyond those associated with aging (Appendix A). FITC-dextran was orally administered to young (8-week-old) and aged (92-week-old) mice, and serum levels of LPS and FITC-dextran were measured. As shown in Figure 2A, aged mice had significantly higher serum levels of both LPS and FITC-dextran than young mice, indicating increased intestinal permeability. Elevated serum IL-6 levels in aged mice further suggested systemic inflammation. Histological analysis of colonic tissues using hematoxylin and eosin (H&E) staining revealed structural damage in aged mice (Figure 2B). Examination of tight junction proteins revealed decreased mRNA expression levels of barrier-forming claudin-1 and claudin-4 in aged mice (Figure 2C), indicating impaired barrier function.

Gut microbiota composition was also analyzed. In young mice, the predominant genera were *Muribaculaceae*, *Lachnospiraceae_NK4A136_group,* and *Lactobacillus*. In aged mice, the relative abundance of these beneficial bacteria decreased (Figure 2D). In particular, aged mice presented significant reductions in the abundances of *Alistipes, Parabacteroides, Clostridiales vadin BB60*, *Lachnoclostridium*, and *ASF356* (Figure 2E). Previous studies have reported inverse correlations between the abundance of these bacteria and the incidence or severity of inflammatory bowel disease and obesity [31,32,33]. Therefore, the observed changes suggest that aging negatively affects the gut microbial environment.

### 2.5. Ulvan-Na Administration Restores the Intestinal Barrier in Mice with Aging-Dependent Leaky Gut

Next, it was investigated whether oral administration of ulvan or ulvan-Na could ameliorate leaky gut symptoms in aged mice. Based on a previous report suggesting the physiological activity of ulvan at a dose of 50 mg/kg, it was decided to administer 50 mg/kg of ulvan to the mice in this experiment as well [34]. The mice were orally administered ulvan or ulvan-Na at 50 mg/kg daily for 25 days. No significant changes in body weights were observed during the treatment period (Figure 3A). Colon shortening is a common indicator of colon dysfunction. Compared with the control-treated group, the ulvan-Na-treated group presented significant increases in colon lengths, suggesting improved gut health (Figure 3B). Ulvan treatment had no effect on colon lengths. As shown in Figure 3C,D, the serum levels of LPS and FITC-dextran were significantly reduced only in the ulvan-Na-treated group. Elevated serum LPS levels in aged mice may cause damage to various organs, including the liver. Measurement of serum alanine aminotransferase (ALT) activity, a marker of liver injury, revealed that ulvan-Na administration significantly reduced ALT levels, suggesting a correlation between improved liver function and enhanced intestinal barrier integrity (Figure 3E). Ulvan treatment did not have similar effects. In addition, serum IL-6 levels were reduced following the administration of ulvan-Na (Figure 3F). Histological examination revealed that ulvan-Na treatment ameliorated the mild colonic damage observed in aged mice (Figure 3G), whereas ulvan had no effect. These results suggest that ulvan-Na ameliorates age-related leaky gut by restoring intestinal barrier function.

Changes in the gut microbiota before and after treatment were also analyzed. Ulvan administration led to an increase in abundance of beneficial *Clostridiales vadin BB60*, but decreased abundance of *[Eubacterium] xylanophilum*, making it difficult to determine overall improvement (Figure 3H). In contrast, ulvan-Na treatment increased the abundance of *Clostridiales vadin BB60* but decreased the abundance of *Turicibacter*, a genus associated with intestinal inflammation [35] (Figure 3I). These findings suggest that ulvan-Na positively modulates the gut microbiota and contributes to improved gut health.

### 2.6. Ulvan-Na Enhances Tight Junction Function by Upregulating Claudin-1 Expression

To elucidate the molecular mechanisms underlying the improved barrier function observed in mice following ulvan-Na treatment, the mRNA expression of tight junction components in intestinal tissues was examined. As shown in Figure 4A, ulvan-Na administration increased the expression of barrier-forming claudin-1 and decreased the expression of channel-forming claudin-2. Ulvan treatment did not affect the mRNA expression of these proteins. The expression of the mucin-related genes Muc2 and Lypd8 remained unchanged with both treatments. Immunohistochemical analysis confirmed increased claudin-1 expression in the intestinal epithelia of the ulvan-Na-treated mice, suggesting enhanced tight junction formation (Figure 4B). Ulvan treatment did not produce this effect. In addition, both the ulvan and ulvan-Na treatments increased the protein expression of β-catenin in intestinal tissues (Figure 4C), which is critical for maintaining normal intestinal tissue turnover. These findings suggest that ulvan-Na may promote tissue repair by upregulating β-catenin expression.

### 2.7. Ulvan-Na Suppresses LPS-Induced Barrier Dysfunction

Given that tight junction integrity can be compromised by LPS-induced inflammatory responses [36], the potential of ulvan-Na to protect against such effects was investigated. Using human induced pluripotent stem cell (iPSC)-derived intestinal epithelial (F-hiPSC8) cells expressing the LPS receptor TLR4, transepithelial electrical resistance (TEER) was assessed as a measure of barrier function [37]. LPS treatment significantly reduced the TEER values, indicating decreased barrier integrity (Figure 5A). Cotreatment with ulvan partially attenuated this reduction, whereas treatment with ulvan-Na almost completely prevented it. The mRNA expression of the inflammatory cytokines IL-1β and IL-6 was also measured. Cotreatment with ulvan-Na suppressed the LPS-induced upregulation of these cytokines, whereas ulvan did not (Figure 5B). In RAW264 macrophages, ulvan-Na treatment inhibited the LPS-induced expression of inflammatory cytokines, whereas ulvan failed to do so and even increased IL-6 expression (Figure 5C,D). Since the change in treatment from ulvan to ulvan-Na changes the functionality of the cells, we considered the possibility that there is a difference in the physical properties of ulvan and ulvan-Na, examined their water absorbencies, and found that ulvan-Na has relatively greater water absorbency. These results suggest that ulvan-Na treatment suppresses LPS-induced inflammation and preserves barrier function, whereas ulvan might have proinflammatory effects.

### 2.8. Ulvan-Na Upregulates Claudin-1 Expression to Improve Barrier Function

To determine whether ulvan-Na treatment could improve tight junction function independent of inflammatory pathways, we used Caco-2 intestinal epithelial cells, which have low TLR4 expression [38]. Ulvan and ulvan-Na treatments did not affect the growth rates or inflammatory cytokine expression of these cells (Appendix A). TEER measurements revealed that ulvan-Na treatment accelerated tight junction formation in Caco-2 cells compared with that in control and ulvan-treated cells (Figure 6A). Ulvan-Na treatment increased claudin-1 mRNA and protein expression while decreasing claudin-2 mRNA expression (Figure 6B,C). Immunofluorescence staining confirmed the increased localization of claudin-1 at cell–cell junctions with ulvan-Na treatment; however, the localization of claudin-2 was unclear (Figure 6D). These results indicate that ulvan-Na stimulates barrier formation and tight junction function via increased claudin-1 localization between cells.

### 2.9. Ulvan-Na Promotes the Nuclear Translocation of β-Catenin to Increase Claudin-1 Expression

The molecular mechanism by which ulvan-Na induces claudin-1 expression was investigated. Since β-catenin is known to regulate claudin-1 transcription [39], its expression and localization were examined. Ulvan-Na treatment increased β-catenin protein expression and promoted its nuclear translocation without affecting the phosphorylation of GSK-3β or β-catenin, suggesting that this activation of β-catenin was independent of the canonical Wnt pathway (Figure 7A–C). Inhibition of β-catenin transcriptional activity by FH535 [40] treatment attenuated the ulvan-Na-induced increases in TEER and claudin-1 expression (Figure 7D–F). These results suggest that ulvan-Na treatment enhances barrier function by promoting β-catenin-mediated transcription of claudin-1.

## 3. Discussion

In this study, an outdoor mass aquaculture method for *U. meridionalis* was established to produce ulvan-Na by modifying the cation content of the water-soluble polysaccharide ulvan. The development of ulvan-Na can increase the potential of ulvan as a health food ingredient to improve age-related leaky gut. Furthermore, ulvan-Na was found to activate mitochondria in aging cells, improve the gut microbiota in aged mice, and restore intestinal barrier function. Specifically, the mechanism by which ulvan-Na promotes the expression and intercellular localization of claudin-1 through activation of β-catenin was elucidated. Therefore, this research suggests that ulvan-Na is a sustainable material with novel bioactivities that enhance the physiological functions of ulvan. The significance of various cations, along with potential differences in the physical and chemical properties and structures, particularly in relation to sodium content, has been identified as an important subject for future research.

First, it was demonstrated that mass aquaculture of *U. meridionalis* is possible using the germling cluster method in land-based aquaculture. This method suspends seaweed as spherical clusters in tanks, maximizing the seaweed weight per unit volume, and making it the most efficient land-based aquaculture technique [21]. In previous studies, *U. meridionalis* exhibited a daily growth rate of approximately fourfold under controlled laboratory conditions at a seawater temperature of 30 °C [22]. In contrast, under the outdoor aquaculture conditions of this study, the daily growth rate was approximately 2.5-fold; though lower than that observed in the laboratory, this is greater than the growth rate of *U. prolifera*, which responds to an established outdoor cultivation method with a daily growth rate of approximately 1.4-fold [21]. In addition, common herbaceous plants such as grasses or rapeseed have daily growth rates ranging from 1.01 to 1.03-fold [41], and even bamboo, which is considered one of the fastest-growing terrestrial plants, has an average daily growth rate of approximately 1.1-fold [42]. Therefore, *U. meridionalis* grows remarkably faster than terrestrial plants do, and its production through land-based aquaculture is a useful and highly efficient method for biomass resource production.

Next, it was clarified that ion exchange treatment alters the functionality of ulvan derived from *U. meridionalis*. Conventional ulvan did not have anti-aging effects; however, ulvan with a modified cation composition by ion exchange, particularly ulvan-Na, suppressed senescence-associated β-galactosidase activity and increased ATP and collagen production in human fibroblasts decreased by aging. Furthermore, it was suggested that the proinflammatory effect of ulvan disappears when the cations contained therein are replaced by sodium. Given that both ulvan and ulvan-Na have molecular weights of approximately 800 kDa, it is unlikely that they are absorbed into cells, suggesting that they may exert effects through interaction with TLR4, the receptor for LPS. Notably, ulvan-Na is expected to inhibit the interaction between LPS and TLR4 without activating TLR4, as it suppresses LPS-induced inflammatory responses. Additionally, the affinity of ulvan-Na for water is greater than that of ulvan, potentially leading to changes in its higher-order structure when it is solubilized, which may alter its interaction with TLR4 and its cellular effects; however, detailed analyses are needed in future studies.

With respect to the leaky gut model using aged mice, previous reports have revealed changes in the gut microbiota and tissue morphology [43]. Although the two experiments used mice of different ages, both sets of mice displayed characteristic features of leaky gut and could, therefore, be regarded as models of age-dependent leaky gut. In the initial experiment, older mice were used to determine whether aging was associated with intestinal barrier dysfunction and to identify reliable leaky gut indicators. Once these indicators had been established, we confirmed that they were also altered in 74-week-old mice. Based on this finding, 74-week-old mice were chosen for evaluation of ulvan-Na, as this age allows an aging-related leaky gut model to be established within a shorter experimental period while retaining the relevant pathophysiological characteristics. In in vivo experiments using age-dependent leaky gut model mice, the oral administration of ulvan-Na restored intestinal epithelial barrier function by increasing claudin-1 expression and reducing the serum levels of LPS and inflammatory cytokines. Subsequent in vitro experiments suggested that ulvan-Na suppresses LPS-induced inflammatory responses and acts directly on cells to increase β-catenin expression, promote its nuclear localization, and subsequently increase the expression level and intercellular localization of claudin-1. Previous studies have shown that the activation of β-catenin is primarily mediated by the Wnt-GSK3β pathway [44]; however, ulvan-Na does not affect the phosphorylation state of GSK3β, suggesting that this pathway is not activated. The observed increase in total β-catenin expression suggests that a stabilized fraction of β-catenin translocates to the nucleus to mediate downstream signaling.

In aged mice, the abundance of *Clostridiales vadin BB60*, which significantly decreased, was significantly increased by ulvan-Na administration. These findings suggest that ulvan-Na promotes the proliferation of beneficial bacteria that have been reduced by host aging. The abundance of the *Clostridiales vadin BB60* clade is inversely correlated with obesity, dyslipidemia, and insulin resistance in mouse models and with BMI, weight, and waist circumference in women [45,46]. Therefore, increasing the abundance of these gut bacteria is thought to maintain or improve the gut environment. Since ulvan-Na administration reduces the abundance of *Turicibacter,* which exacerbates gut inflammation and worsens the gut environment [47], ulvan-Na not only results in the gut microbiota being in a relatively young state but also shifts the gut environment toward a healthier state. In this study, we verified that ulvan-Na administration altered the composition of the gut microbiota. However, we could not determine the direct impact of specific microbial groups on the intestinal environment. This limitation is an important topic for future research. These results suggest that ulvan-Na may improve age-related leaky gut both structurally and functionally by improving intestinal barrier function and modulating the gut microbiota.

While this study demonstrated the anti-aging effects of ulvan-Na and its ability to improve intestinal barrier function, its molecular mechanisms need to be completely elucidated. In particular, further investigations are needed to understand the detailed activation mechanisms of the β-catenin pathway and its effects on other signaling pathways. Future studies should focus on detailed molecular mechanism analyses and safety assessments to advance the clinical application of ulvan-Na.

Previous studies have described various physiological activities of ulvan; however, its functionality as a health food has not been fully explored. In addition, because the functionality of ulvan is known to vary significantly depending on species differences and growth environments, technological development is needed for mass culture of specific species under stable conditions.

## 4. Materials and Methods

### 4.1. Strains and Seedling Stock Preparation

*U. meridionalis* strain E16, preserved as a unialgal isolate at the Usa Marine Biological Institute, Kochi University, was utilized [22]. The seedling stocks used for the growth experiments were prepared using the “germling cluster” technique, which is suitable for free-floating macroalgal cultivation. A 12-h light/dark cycle (L12:12) at an intensity of 100–200 µmol photons m^−2^ s^−1^ was maintained for *U. meridionalis.* Once they reached a length of 5 mm or greater, cultivation proceeded outdoors in tanks. The seedlings were gradually scaled up based on their growth stages and finally cultivated in 1-ton outdoor tanks. These tanks were fed UV-sterilized surface seawater pumped from Usa Bay in Kochi Prefecture, Japan. Nutrients, specifically nitrogen and phosphorus, were added at a rate of 1 mL/min (concentrations: N: 0.5 mM, *p*: 0.05 mM), and aeration was provided to ensure adequate mixing of *U. meridionalis.*

### 4.2. Extraction of Ulvan and Exchange of the Cations Contained in Ulvan

The well-washed U. meridionalis was dried and powdered. The powder was treated with hot water at 90 °C, and then the supernatant was precipitated with ethanol. The extracted ulvan was dried and crushed. The purity of ulvan is greater than 90%. Elemental analysis of ulvan yielded C/N/S ratios of 30/0.4/6. In contrast, ion chromatography suggested that salt contamination was approximately 1%. Therefore, the proportion of impurities in ulvan was estimated to be 5–10%. The extracted ulvan (2.5 g) was dissolved in 500 mL of distilled water. The 0.5% ulvan aqueous solution (500 mL) was treated 5 times with 0.5 g of an OH-form strong-base anion exchange resin (Amberlite IRA410 OH, Organo, Japan) at room temperature for 1 h to remove free sulfate and chloride ions contaminating the aqueous solution. The ulvan-Na samples were produced by the ion exchange method using a Na+-form strong-acid cation exchange resin (Amberlite IR120B Na, Organo, Japan) at room temperature for 1 h. Ulvan-Na (41 mol%) and ulvan-Na (93 mol%) were obtained by the methods using 0.3 g and 1.0 g of Amberlite IR120B Na resin, respectively, from the treated ulvan aqueous solution (100 mL). Ulvan-H (50 mol%) was produced by exchanging the Na-ulvan (41 mol%) aqueous solution (100 mL) using an H+-form strong-acid cation exchange resin (Amberlite IR120B H, Organo, Japan) (0.5 g) at room temperature for 1 h. After the ion exchange treatments, the supernatants were lyophilized. The obtained samples were analyzed by gel filtration chromatography (GPC8320, Tosoh, Japan) with a Shodex OHpak SB-804 HQ column (Showa Denko, Japan), an inductively coupled plasma–atomic emission spectroscopy (ICP–AES) analyzer (ICPE 9000, Shimadzu, Japan), ion chromatography (CDD-10A vp, Shimadzu, Japan) with a TSKgel Super IC-A/C column (Tosoh, Japan), and a total organic carbon analyzer (TOC-N, Shimadzu, Japan). A flow diagram of the process was shown in Figure 8.

### 4.3. Cell Lines

NB1RGB (human skin fibroblasts, provided by Riken BRC, Tokyo, Japan) cells were cultured in MEMα medium (Wako, Tokyo, Japan) supplemented with 10% FBS, 100 U/mL penicillin, and 100 µg/mL streptomycin. The cell cultures were passaged at a 1:4 ratio every three days and maintained at 37 °C in a 5% CO_2_ atmosphere. Each cell line was categorized into one of two types on the basis of days in culture: NB1RGB young cells (8–20 days) and aging cells (60–70 days) [27]. The same cell line used by Machihara et al. [27] was also employed in the present study.

### 4.4. Determination of SA-β-gal Activity

Cellular SA-β-gal activity was measured using a senescence detection kit (#ab65351, Abcam, Cambridge, UK), following the manufacturer’s instructions. Images were captured with a microscope (IX73, Olympus, Tokyo, Japan) and processed with Adobe Photoshop Elements 2023.

### 4.5. ATP Assay

Intracellular ATP levels were determined via a CellTiter-Glo 2.0 assay kit (#G9241, Promega, Madison, WI, USA) according to the manufacturer’s protocol. ATP production values were normalized to the total protein content.

### 4.6. Collagen Assay

Collagen levels were measured using a collagen quantitation kit (#COL-001, Cosmo Bio, Tokyo, Japan) in accordance with the manufacturer’s instructions. Collagen production was normalized to the total protein content. Fluorescence was detected using a fluorescence microplate reader (Infinite M200, Tecan, Tokyo, Japan) with a 360/485-nm filter pair.

### 4.7. Real-Time Quantitative PCR (qRT–PCR)

qRT–PCR was performed as previously described [48]. The total RNA expression was normalized in each reaction using β-actin cDNA as an internal standard. The primers used are listed in Appendix A.

### 4.8. Immunoblotting Analysis

Immunoblotting experiments were conducted as previously described [48]. The antibodies used for immunoblotting were specific to the following proteins: claudin-1, claudin-2, *p*-β-catenin, β-catenin, p-GSK3β, GSK3β, and SOD2 (Cell Signaling); and β-actin (Sigma). The antibodies were diluted at a 1:1000 ratio, except for the anti-β-actin antibody (1:10,000 dilution). The secondary antibodies used were purchased from Promega (anti-rabbit and anti-mouse, used at a 1:5000 dilution).

### 4.9. Mouse Model of Aging-Dependent Leaky Gut

All experimental protocols and surgical procedures were approved by the Animal Research Committee of Kochi University (Permit Numbers: P-00064, Q-00026, R-00041). Animal experiments were conducted according to the guidelines for animal experimentation of Kochi University and the recommendations of the ARRIVE guidelines.

The mice were housed in a room at 24 ± 3 °C and 55 ± 10% relative humidity with a 12-h light-dark cycle, and food and water were provided ad libitum. C57BL/6 male mice (8 (*n* = 5), 80 (*n* = 5), 92 (*n* = 5), and 100 (*n* = 5) weeks of age) were evaluated for their intestinal status. C57BL/6 male mice (74 weeks) were orally administered 50 mg/kg ulvan or ulvan-Na daily for 25 days, and body weights were measured daily. After completion of oral administration, the animals were fasted for 4 h. Two hours after oral administration of 200 mg/kg FITC-dextran (4 kDa), blood samples were collected. The colons and livers were then collected, and the lengths of the colons and the weights of the livers and kidneys were measured. The colon tissues were fixed in 4% PFA for paraffin sectioning, and the remaining tissues were used for mRNA analysis. Following paraffin embedding, sections of liver and kidney tissues were cut, stained with H&E, and mounted (PathoMount; Wako, Tokyo, Japan) prior to microscopic examination using a microscope (BZ-X800, KEYENCE, Osaka, Japan). For immunofluorescence staining, the sections were incubated with anti-claudin-1, anti-claudin-2, or anti-β-catenin antibodies and subsequently treated with FITC- or Alexa 488-conjugated goat anti-rabbit immunoglobulins (Wako, Tokyo, Japan). Afterward, the sections were stained with DAPI. The samples were mounted with Fluomount (Sigma–Aldrich, St. Louis, MO, USA), examined under a microscope (BZ-X800, KEYENCE, Osaka, Japan), and processed using Adobe Photoshop software.

### 4.10. Mature Serum LPS, FITC-Dextran, IL-6 Levels, and Serum ALT Activity

Serum LPS (#A39552; Thermo Fisher Scientific, Waltham, MA, USA), serum IL-6 (#M6000B; R&E Systems, Minneapolis, MN, USA), and serum ALT activity (#700260; Cayman Chemical, Ann Arbor, MI, USA) were measured using commercially available enzymatic kits according to the manufacturer’s protocols. Serum FITC-dextran concentrations were determined with a fluorescence microplate reader (Infinite M200, TECAN, Tokyo, Japan) using filter pairs of 485 nm/535 nm.

### 4.11. Microbial Community Analysis

Total microbial DNA from the fecal samples (200 mg) of the mice was extracted using the DNeasy Powersoil Kit (Qiagen, Hilden, Germany), following the manufacturer’s instructions. Then, 16S rRNA gene amplification, sequencing, and sequencing data analysis were performed by Azenta Life Sciences (Tokyo, Japan).

### 4.12. Immunofluorescence Staining of Cultured Cells

The cells were fixed with ice-cold acetone or 4% PFA, blocked with 2.5% goat serum for 10 min, incubated for 12 h with anti-claudin-1, anti-claudin-2, or total β-catenin antibody in 3% bovine serum albumin, and subsequently incubated for 1 h with FITC- or Alexa 488-conjugated goat anti-rabbit immunoglobulins (Wako, Tokyo, Japan). Afterward, the sections were stained with DAPI. The samples were mounted with Fluoromount (Sigma-Aldrich, St. Louis, MO, USA), examined under a microscope (BZ-X800, KEYENCE, Osaka, Japan), and processed using Adobe Photoshop software.

### 4.13. TEER Measurement

The cells were cultured on permeable membrane inserts (Transwell^®^; Corning, Coster, NY, USA) until a confluent monolayer formed. TEER values were determined using an epithelial volt-ohm meter (Millicell^®^ ERS-2; Merck, Burlington, MA, USA) equipped with chopstick electrodes. The TEER value was calculated by subtracting the resistance of a blank insert (insert without cells) from the measured resistance of the cell-covered insert. The resulting resistance (in ohms) was multiplied by the effective membrane surface area (cm^2^) to express the TEER in standard units (Ω·cm^2^).

### 4.14. Cell Count Assay

The cells were collected, reconstituted in fresh growth medium, and treated with 0.4% Trypan blue solution (Sigma, St. Louis, MO, USA) at a 1:1 ratio. The stained cell mixture was transferred onto a hemocytometer and observed using a light microscope. Both viable and nonviable cells were counted, and the cell viability was determined as the proportion of live cells relative to the total cell count, expressed as a percentage.

### 4.15. Water Absorption Test

Each sample was measured to 200 mg on a mesh T-bag made to a certain size, and the overall weight was measured (Wdry). Each sample was fully submerged in distilled water for 3 h and then removed using tweezers. Excess water was allowed to drip for 30 s, and surface moisture was gently blotted with filter paper. The wet weight (Wwet) was immediately measured. Water absorption was calculated as follows: Water absorption (g) = Wwet − Wdry.

### 4.16. Statistical Analyses

To evaluate differences between two groups, unpaired data were analyzed using either Student’s t test or Welch’s t test, depending on whether the sample sizes or variances between the groups were equal. For comparisons involving more than two compounds tested on a single cell type, ANOVA was applied. When assessing two cell types treated with multiple compounds, two-way ANOVA was conducted, followed by the Tukey HSD test for post hoc analysis. Statistical significance was defined as a *p* value < 0.05. All statistical analyses were conducted using Mac statistical analysis software (Esumi Co., Tokyo, Japan).

## 5. Conclusions

This study demonstrated that mass aquaculture of *U. meridionalis* through land-based aquaculture is feasible and that ulvan-Na, which possesses anti-aging effects and improves intestinal barrier function, can be obtained through ion exchange treatment of the extracted ulvan. These findings, although limited to results from animal experiments, highlight the potential of *U. meridionalis*-derived ulvan-Na as a functional food material suitable for dietary implementation, thereby contributing to the effective utilization of seaweed resources and supporting the realization of a healthy and long-lived society.

## Figures and Tables

**Figure 1 marinedrugs-23-00390-f001:**
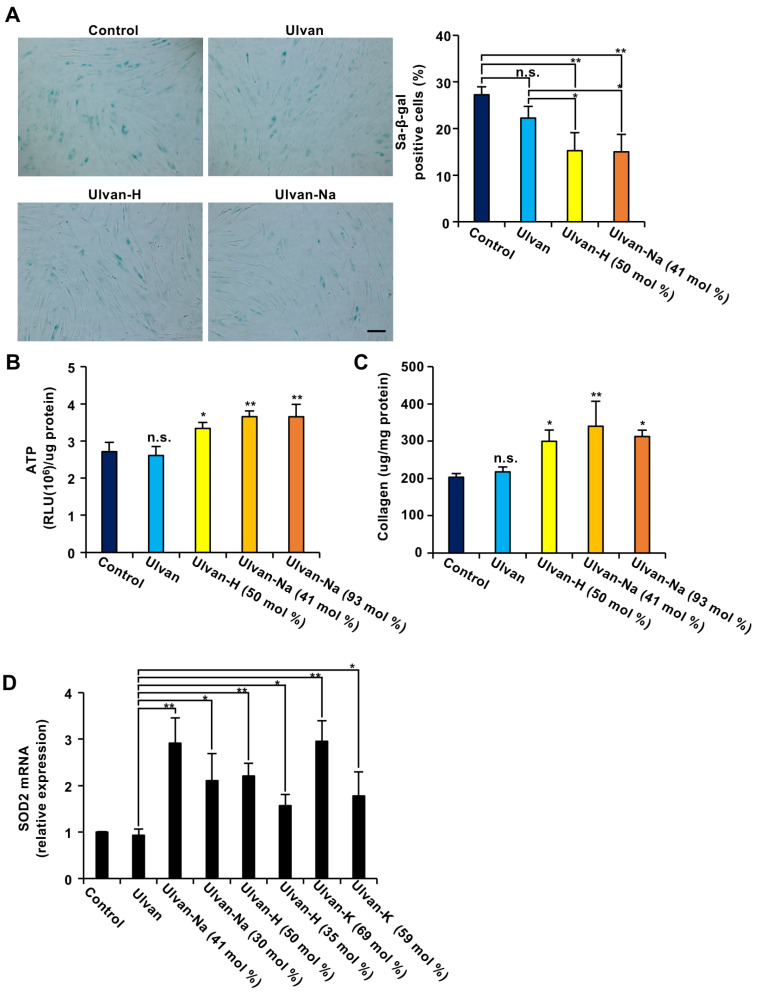
The exchange of ulvan-containing cations with alkali metal ions (AMIs) enhances the antiaging effect of ulvan in aging cells. (**A**) Ulvan-AMI treatment decreased the number of SA-β-gal-positive cells. Aging NB1RGB cells were treated with 100 μg/mL of the indicated polysaccharides for 48 h and subjected to SA-β-gal staining. Images were taken at 10× magnification (scale bar, 50 μm) (left panel), 100–150 cells were counted, and the percentage of SA-β-gal-positive cells among them was determined (*n* = 3) (right panel). (**B**–**D**) Aging NB1RGB cells were treated with 100 μg/mL of the indicated polysaccharides for 24 h (**D**) or 48 h (**B**,**C**). ATP content (**B**) and collagen content (**C**) were then measured in these cells. Collagen production and ATP levels were normalized to the total amount of protein. (**D**) The cells were lysed, and their RNA was then subjected to qPCR. The data are presented as the means ± SDs of three simultaneous experiments performed using three wells on the same plate (**A**) or three wells on separate plates (**B**–**D**). *p* values were calculated using ANOVA following Tukey’s HSD test (**A**–**D**); n.s.: not significant, * *p* < 0.05, ** *p* < 0.01.

**Figure 2 marinedrugs-23-00390-f002:**
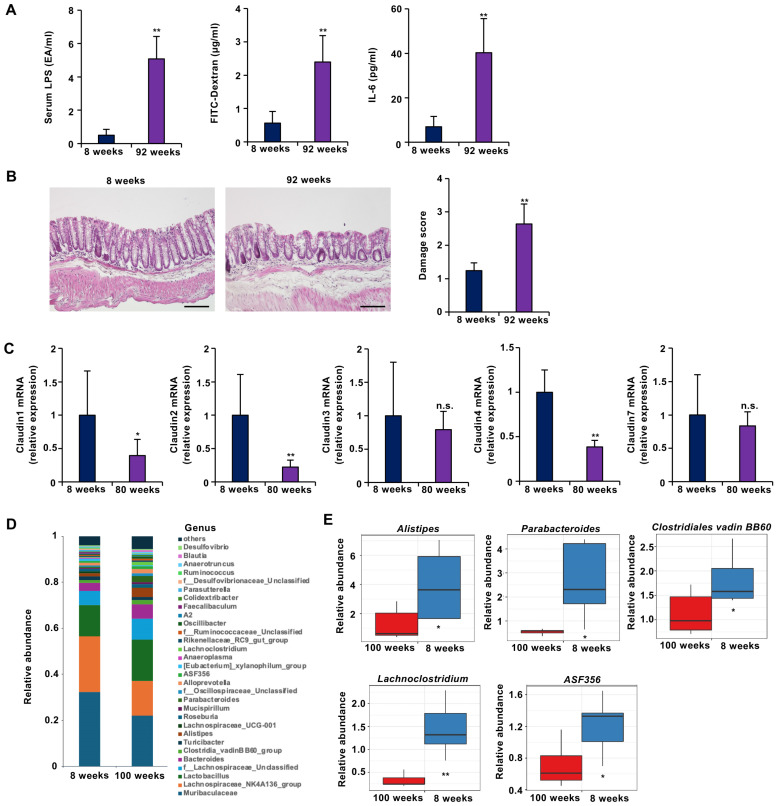
Aged mice presented with leaky gut-like symptoms. (**A**–**C**) Mice aged 8 weeks (young) (*n* = 5) or 92 weeks (aged) (*n* = 5) were subjected to the indicated experiments. (**A**) Intestinal barrier function is disrupted in aged mice. Compared with young mice, aged mice presented significant increases in 4 kDa FITC-dextran (left panel) and endotoxin (LPS) (center panel) leakage from the gut to the blood, as well as serum IL-6 concentrations (right panel). (**B**) Sections of colonic tissues from young and aged mice were prepared and subjected to histological examination (H&E staining), and the damage score and extent of lesions in five independent sections were determined. (**C**) Barrier function-related claudin-1 and claudin-4 mRNA expression was decreased in aging mice (*n* = 5) compared to young mice (*n* = 5). The colonic tissues were removed, and total RNA was extracted. The samples were subjected to real-time quantitative PCR, and the results are expressed relative to those of the control samples (i.e., young mice). (**D**,**E**) The number of beneficial intestinal bacteria decreased with age in the mice. (**D**) Taxonomic profiles of bacterial communities at the genus level (**D**) and the relative abundance of the indicated bacteria (**E**) in all fecal samples collected from young (8-week-old) (*n* = 5) and aged (100-week-old) (*n* = 5) mice. The data are presented as the means ± SDs (**A**–**C**,**E**). *p*-values were calculated using Welch’s t test (**A**–**C**,**E**); n.s.: not significant, * *p* < 0.05, ** *p* < 0.01.

**Figure 3 marinedrugs-23-00390-f003:**
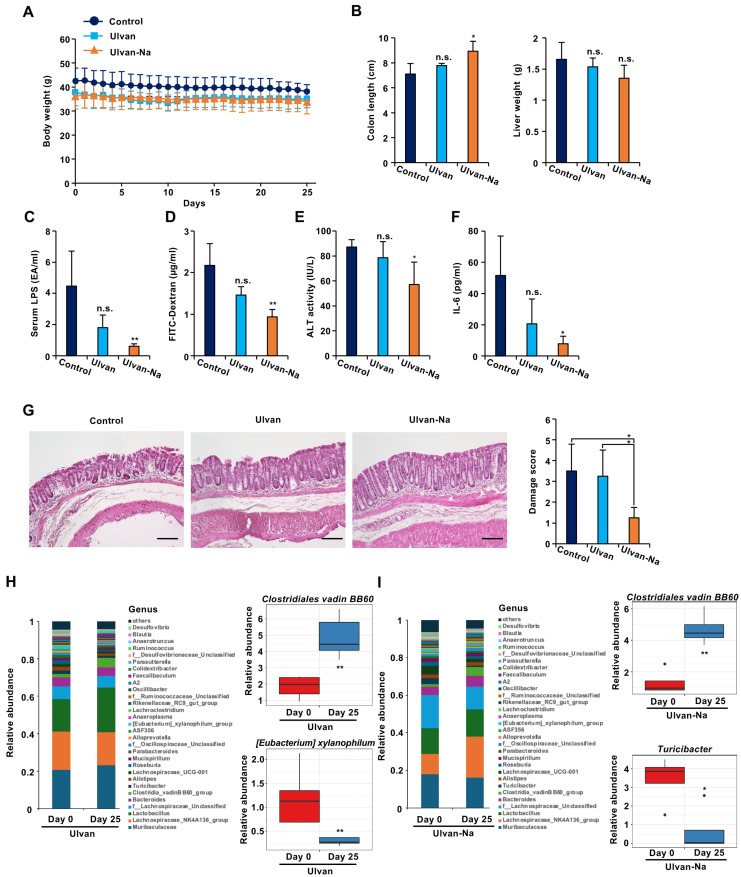
Ulvan-Na treatment ameliorated leaky gut-like symptoms by improving intestinal barrier function and the intestinal microflora. (**A**–**F**) Mice aged 74 weeks were orally administered DW (*n* = 5), 50 mg/kg ulvan (*n* = 4) or 50 mg/kg ulvan-Na (*n* = 4) daily for 25 days. Body weights ((**B**): **left panel**), liver weights ((**B**): **right panel**), serum LPS levels (**C**), serum ALT activity (**E**), and IL-6 concentrations (**F**) were measured. (**D**) After treatment with the indicated compounds, the mice were orally administered 4 kDa FITC-conjugated dextran (200 mg/kg); the serum was subsequently collected, and FITC fluorescence was measured. (**G**) Tissue sections treated with the indicated compounds were prepared and subjected to histological examination (H&E staining), and the damage score and extent of lesions in four independent sections were determined. (**H**,**I**) Administration of ulvan-Na restored the intestinal microbiota toward that of young mice. Taxonomic profiles of bacterial communities at the genus level and the relative abundances of the indicated bacteria in all fecal samples collected from ulvan (**H**)- and ulvan-Na (**I**)-treated mice at days 0 and 25. The data are presented as the means ± SDs (**A**–**I**). *p* values were calculated using ANOVA following Tukey’s HSD test (**A**–**G**) or Welch’s t test (**H**,**I**); n.s.: not significant, * *p* < 0.05, ** *p* < 0.01.

**Figure 4 marinedrugs-23-00390-f004:**
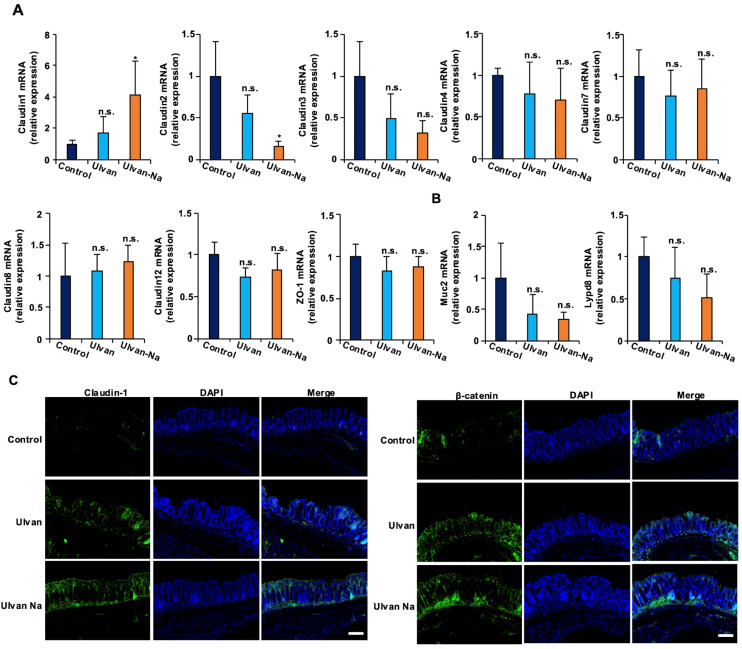
Ulvan-Na treatment increased the expression levels of claudin-1 and β-catenin in the colons of aging mice. (**A**–**C**) Mice aged 74 weeks were orally administered DW (*n* = 4), 50 mg/kg ulvan (*n* = 4), or 50 mg/kg ulvan-Na (*n* = 4) daily for 25 days. The colonic tissues were removed, and total RNA was subjected to real-time quantitative PCR (**A**), or the tissues were subjected to immunofluorescence staining using the indicated primary antibodies (**B**). *p* values were calculated via ANOVA following the Tukey-HSD test (**A**); n.s.: not significant, * *p* < 0.05.

**Figure 5 marinedrugs-23-00390-f005:**
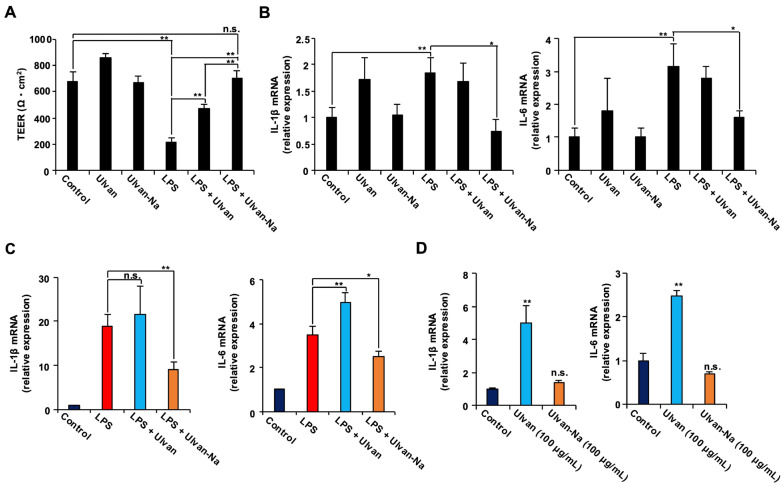
Ulvan-Na treatment suppressed the LPS-induced disruption of tight junctions via inhibition of the inflammatory response in intestinal epithelial cells differentiated from iPS cells. (**A**,**B**) Intestinal epithelial cells differentiated from human iPS cells were cultured on inserts to establish cell polarity and form tight junctions. These cells were subsequently treated with or without 1 µg/mL LPS, 100 µg/mL ulvan, or 100 µg/mL ulvan-Na for 24 h; then, the TEER (**A**) was measured, and isolated RNA was subjected to qPCR (**B**). (**C**,**D**) Ulvan-Na suppressed LPS-induced inflammation, and ulvan-Na lost its proinflammatory effect on RAW264 cells. RAW264 cells were treated with or without 1 µg/mL LPS, 100 µg/mL ulvan, or 100 µg/mL ulvan-Na for 24 h; isolated RNA was then subjected to qPCR. The data are presented as the means ± SDs of three simultaneous experiments performed using three wells on the same plate (**A**,**B**) or three wells on separate plates (**C**,**D**). *p* values were calculated via ANOVA following the Tukey-HSD test (**A**–**D**); n.s.: not significant, * *p* < 0.05, ** *p* < 0.01.

**Figure 6 marinedrugs-23-00390-f006:**
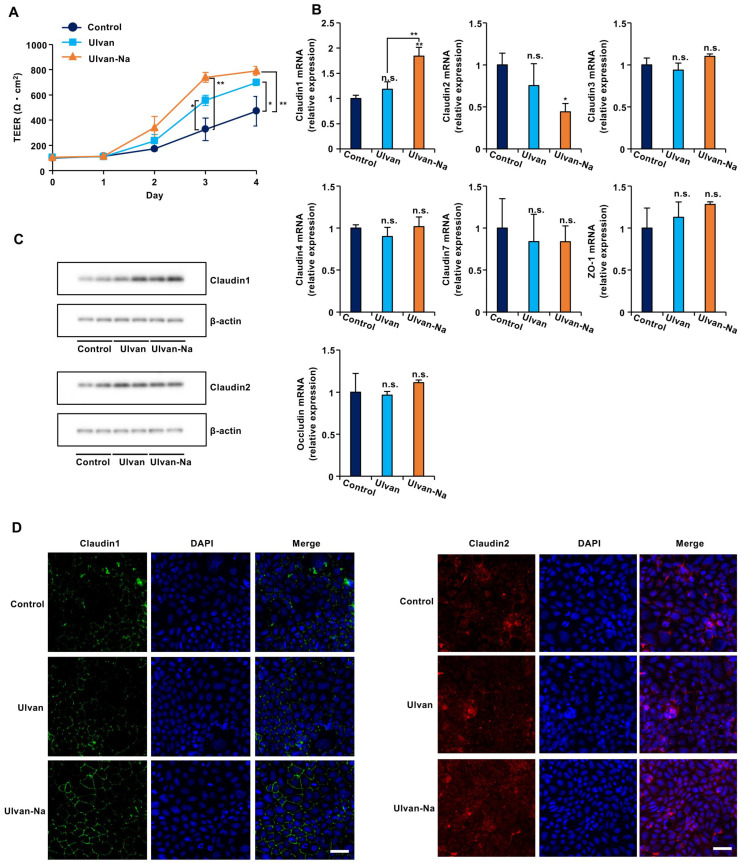
Ulvan-Na treatment enhanced intestinal epithelial cell barrier function by increasing the plasma membrane localization of claudin-1 in Caco-2 cells. (**A**–**C**) Caco-2 cells were cultured on inserts for 24 h (day 0), and these cells were treated with 100 µg/mL of the indicated materials. Then, the TEER was measured on the indicated day (**A**), isolated RNA from the cells was subjected to qPCR (**B**), and cellular proteins were subjected to immunoblotting (**C**) after 3 days. (**D**) Caco-2 cells were treated with 100 µg/mL of the indicated materials for 72 h. The localization of claudin-1 in the cells was determined via an immunofluorescence staining assay using the indicated antibody. These images were taken at 20× magnification (scale bar, 50 μm). *p* values were calculated via ANOVA following the Tukey-HSD test (**A**,**B**); n.s.: not significant, * *p* < 0.05, ** *p* < 0.01.

**Figure 7 marinedrugs-23-00390-f007:**
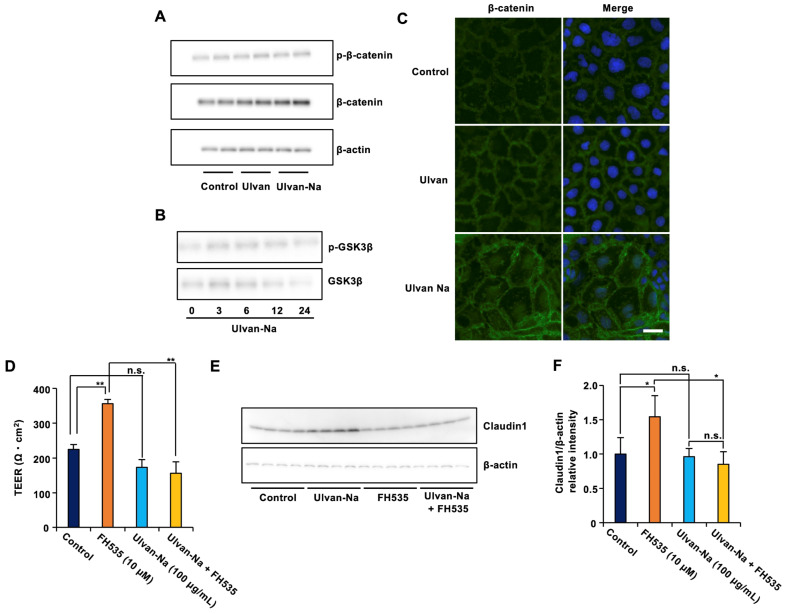
Ulvan-Na treatment promoted intestinal epithelial cell barrier function by increasing β-catenin phosphorylation and nuclear localization. (**A**,**B**) Caco-2 cells were treated with the indicated compounds for 24 h (**A**) or the indicated periods (**B**). The cells were subjected to immunoblotting using the indicated antibodies. (**C**) Caco-2 cells were treated with 100 µg/mL of the indicated materials for 72 h. The localization of β-catenin in the cells was determined via an immunofluorescence staining assay. These images were taken at 20× magnification (scale bar, 20 μm). (**D**–**F**) Caco-2 cells were cultured on inserts for 24 h, and these cells were treated with the indicated materials. Then, the TEER was measured (**D**), and cellular proteins were subjected to immunoblotting (**E**) after 2 days. (**F**) The intensity of the claudin1 (**E**) was determined (the levels of claudin1 was reported relative to those of β-actin). *p* values were calculated via ANOVA following the Tukey-HSD test (**D**,**F**); n.s.: not significant, * *p* < 0.05, ** *p* < 0.01.

**Figure 8 marinedrugs-23-00390-f008:**
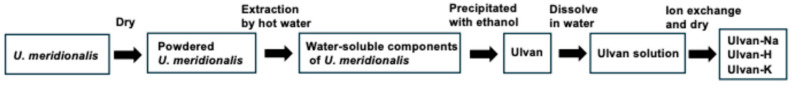
Dagram of ion exchange ulvan preparation.

**Table 1 marinedrugs-23-00390-t001:** Exchange of ulvan-containing cations with sodium ions and protons.

	Molecular Weight/×10^5^	Cation Concentrations/mmol g^−1 (a)^				
		Na^+^	K^+^	Ca^2+^	Mg^2+^	H^+^
Ulvan	8	0.2	0.4	0.2	1.0	trace
Ulvan-H (35 mol%)	8	0.1	0.1	0.2	0.8	0.6
Ulvan-H (50 mol%)	8	0.3	0.0	0.1	0.6	1.1 ^(b)^
Ulvan-Na (30 mol%)	8	0.6	0.2	0.1	0.8	trace
Ulvan-Na (41 mol%)	8	1.0	0.1	0.1	0.7	trace
Ulvan-Na (93 mol%)	8	2.7	0.0	0.1	0.0	trace
Ulvan-K (59 mol%)	8	0.1	1.6	0.1	0.5	trace
Ulvan-K (69 mol%)	8	0.0	1.8	0.1	0.3	trace

^(a)^ The metal ion concentrations were determined via ICP–AES. ^(b)^ The H^+^ concentration of Ulvan-H (50 mol%) was estimated by subtracting the total positive value of the metal ions in Ulvan-H (50 mol%) from the average value of the other ulvan compounds.

## Data Availability

All data needed to evaluate the conclusions in this paper are presented in the paper and/or the Appendix A. Additional data related to this paper may be requested from the author.

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
