# Peer review of "Ulvan-Na, an Ulvan Subjected to Na+ Cation Exchange, Improves Intestinal Barrier Function in Age-Related Leaky Gut"

_marinedrugs, 2025, doi:10.3390/md23100390_

Round 1

Reviewer 1 Report

Comments and Suggestions for Authors

The manuscript, entitled "Ulvan-Na, an ulvan-polysaccharide subjected to Na+ cation exchange, improves symptoms related to age-related leaky gut," was authored by Maejima et al. This paper investigates the sustainable production and functional development of Ulva meridionalis, a fast-growing seaweed, to improve gut health and mitigate the effects of aging. It demonstrates the feasibility of sustainable U. meridionalis production. These findings underscore the promise of ulvan-Na, a component derived from the filamentous fern, as a novel functional food element with the potential to mitigate age-related intestinal dysfunction. The application of ion exchange treatment resulted in the enhancement of ulvan's functionality, leading to the production of ulvan-Na, which exhibited elevated levels of Na+ and demonstrated superior antiaging properties. While the increase in sodium has been demonstrated to enhance anti-aging effects, the question remains as to whether long-term sodium intake disrupts the body's sodium-potassium balance. This is particularly salient for the elderly, for whom reduced sodium intake is advised. The central question of this study is whether the ingestion of sodium will exceed the recommended daily intake for the human body. The document under review exhibits deficiencies in its design, as well as a fragmented writing style. It has been observed that the sections designated as 2.2–2.5 appear to be isolated, exhibiting a lack of substantial interconnectivity. A more appropriate approach would be to integrate the Results and Discussion sections.

Insert in abstract the most important values which were obtained in this manuscript.

The introduction must be revised. What is the innovation of this paper? The rationale behind the decision to augment the sodium content remains ambiguous.

Table 1: Elucidation of the significance of 50 mol%, 41 mol%, and 93 mol%. An investigation into the presence of discrepancies in the physical and chemical properties and structures of the various samples is warranted, with a particular emphasis on the sodium content. Of particular interest is the variation in sodium content.

Lines 195–206: The author delineates various concentration exchanges; however, only two samples are enumerated in Table 1. The rationale behind the inclusion of multiple inconsistent samples in this context warrants further examination.

A reexamination of the figures is imperative. The legend should be eliminated, and the sample names should be incorporated into the horizontal axis.

The author's study exclusively compared the levels of LPS, FITC dextran, and IL-6 in the blood of mice of varying ages following the oral administration of FIT-dextran. This approach is not sufficiently rigorous. The objective of this study is to determine if there are differences in inflammatory cytokine levels between mice of the same age who did or did not take FIT-dextran orally. The present study seeks to determine whether the observed increase in inflammatory cytokine levels is attributable to the oral administration of FIT dextran or to the patient's age.

Figure 2E is not clear.

In section 2.4, the author compared 8-week-old and 92-week-old mice; however, 74-week-old mice were utilized in the Ulvan treatment. An explanation for this apparent inconsistency is requested.

Line 290: It is imperative that the terms "Clostridiales vadin BB60" and "Lachnospiraceae NK4A136" be italicized.

Line 519: It is imperative to supplement the core parameters for the ion exchange protocol.

Line 537: The definition of the cellular senescence state based exclusively on the number of culture days is inadequate. Replicative senescence is contingent upon validation by at least one or more senescence biomarkers. It is recommended that the data be supplemented with at least one (preferably two) senescence biomarker, such as SA-β-Gal staining, in order to confirm that the cells at 60-70 days have indeed entered a senescent state.

Line 574: It is imperative to specify the number of animals utilized in each experimental group.

Line 573: It is imperative that the complete environmental conditions for mouse rearing be supplied, including but not limited to temperature and humidity.

Line 583: It should be noted that kidney tissues were also collected during the sampling process.

Line 594: A discrepancy exists between the title and the content: The method description utilizes the term "serum triglycerides" for the measurement of bodily substances, while the title makes reference to "IL-6" without providing any further elaboration or context within the main text. A thorough revision of this text is therefore essential.

Line 623: It is imperative that the format of company addresses be standardized. To that end, either detailed addresses must be provided for all companies or only the company names and their respective countries must be included.

It has been observed that the images contained within the file titled "IDR-3869893-original-images" appear to be of suboptimal clarity.

Author Response

The manuscript, entitled "Ulvan-Na, an ulvan-polysaccharide subjected to Na+ cation exchange, improves symptoms related to age-related leaky gut," was authored by Maejima et al. This paper investigates the sustainable production and functional development of Ulva meridionalis, a fast-growing seaweed, to improve gut health and mitigate the effects of aging. It demonstrates the feasibility of sustainable U. meridionalis production. These findings underscore the promise of ulvan-Na, a component derived from the filamentous fern, as a novel functional food element with the potential to mitigate age-related intestinal dysfunction. The application of ion exchange treatment resulted in the enhancement of ulvan's functionality, leading to the production of ulvan-Na, which exhibited elevated levels of Na+ and demonstrated superior antiaging properties. While the increase in sodium has been demonstrated to enhance anti-aging effects, the question remains as to whether long-term sodium intake disrupts the body's sodium-potassium balance. This is particularly salient for the elderly, for whom reduced sodium intake is advised. The central question of this study is whether the ingestion of sodium will exceed the recommended daily intake for the human body. The document under review exhibits deficiencies in its design, as well as a fragmented writing style. It has been observed that the sections designated as 2.2–2.5 appear to be isolated, exhibiting a lack of substantial interconnectivity. A more appropriate approach would be to integrate the Results and Discussion sections.

Response: Thank you for important comments. According to WHO guidelines, adults are recommended to consume less than 2,000 mg of sodium per day. As 1 g of Ulvan-Na (43 mol%) contains 20 mg of sodium, even with a daily intake of 1 g, this represents only about 1/100 of the recommended limit, suggesting that it would not adversely affect health. In addition, as some parts of the Results section included elements of the Discussion, revisions were made to clearly separate the Results and Discussion, taking into account comments from other reviewers.

Comment: Insert in abstract the most important values which were obtained in this manuscript.

Response: Thank you for the suggestion. The abstract has been revised to include the most important values obtained in this study.

Comment: The introduction must be revised. What is the innovation of this paper? The rationale behind the decision to augment the sodium content remains ambiguous.

Response: Thank you for the comment. The key innovation of this study is the development of ulvan-Na, which improves age-related leaky gut, while also emphasizing the importance of environmentally friendly and sustainable raw material production. The simultaneous focus on environmentally conscious production and the development of health-promoting materials represents the innovative aspect of this work. In addition, the focus on sodium content arose from our experimental results; therefore, the Introduction was revised to note that ion-exchanged ulvan has not previously been investigated.

Comment: Table 1: Elucidation of the significance of 50 mol%, 41 mol%, and 93 mol%. An investigation into the presence of discrepancies in the physical and chemical properties and structures of the various samples is warranted, with a particular emphasis on the sodium content. Of particular interest is the variation in sodium content.

Response: Thank you for your important comment. We have described this issue in the discussion section as a topic for future research. At present, we anticipate that ulvan-Na exhibits higher water retention than ulvan, indicating some structural change has occurred. Furthermore, since its physiological activity also differs, we expect its functionality within living organisms to be distinct. Line720-722.

Comment: Lines 195–206: The author delineates various concentration exchanges; however, only two samples are enumerated in Table 1. The rationale behind the inclusion of multiple inconsistent samples in this context warrants further examination.

Response: Thank you for the comment. Table 1 has been revised to ensure consistency with the concentration exchanges described, and the rationale for sample selection has been clarified in the revised text. Line404-406.

Comment: A reexamination of the figures is imperative. The legend should be eliminated, and the sample names should be incorporated into the horizontal axis.

Response: Thank you for the suggestion. The figures have been reexamined, the legends removed, and the sample names incorporated into the horizontal axis.

Comment: The author's study exclusively compared the levels of LPS, FITC dextran, and IL-6 in the blood of mice of varying ages following the oral administration of FIT-dextran. This approach is not sufficiently rigorous. The objective of this study is to determine if there are differences in inflammatory cytokine levels between mice of the same age who did or did not take FIT-dextran orally. The present study seeks to determine whether the observed increase in inflammatory cytokine levels is attributable to the oral administration of FIT dextran or to the patient's age.

Response: The 4 kDa FITC-dextran is commonly used as a marker to evaluate intestinal barrier function in mice. It does not induce inflammation by itself but indicates barrier dysfunction when detected in the blood. Under standard experimental conditions, it is considered safe and non-inflammatory. In experiments with aged mice, serum IL-6 concentrations were measured after oral administration of 4 kDa FITC-dextran. The results showed that 4 kDa FITC-dextran did not enhance IL-6 levels beyond those attributable to aging (supplementary Figure S3 in revised manuscript). Line 444-447.

Comment: Figure 2E is not clear.

Response: Figure 2E has been replaced with a higher-resolution version for improved clarity.

Comment: In section 2.4, the author compared 8-week-old and 92-week-old mice; however, 74-week-old mice were utilized in the Ulvan treatment. An explanation for this apparent inconsistency is requested.

Response: In this study, mice aged 74 to 100 weeks were used as an aged mouse model. All mice exhibited elevated blood LPS concentrations and increased intestinal permeability of 4 kDa FITC-dextran into the bloodstream, confirming the presence of leaky gut. Therefore, these mice were used as an age-dependent leaky gut model. However, no comparisons of the data between mice of different ages were performed. Line 753-784.

Comment: Line 290: It is imperative that the terms "Clostridiales vadin BB60" and "Lachnospiraceae NK4A136" be italicized.

Response: The terms Clostridiales vadin BB60 and Lachnospiraceae NK4A136 have been italicized throughout the revised manuscript.

Comment: Line 519: It is imperative to supplement the core parameters for the ion exchange protocol.

Response: Thank you for the comment. The core parameters for the ion exchange protocol have been supplemented and are described in detail in the Methods section. Line 841-864.

Comment: Line 537: The definition of the cellular senescence state based exclusively on the number of culture days is inadequate. Replicative senescence is contingent upon validation by at least one or more senescence biomarkers. It is recommended that the data be supplemented with at least one (preferably two) senescence biomarker, such as SA-β-Gal staining, in order to confirm that the cells at 60-70 days have indeed entered a senescent state.

Response: The same cell line used by Machihara et al. Aging (Albany NY) 2022, 14, (19), 7662-7691. was also employed in the present study. This paper showed many type of senescence biomarker, such as SA-β-Gal staining, p16 and mitochondrial activity etc. Thus, this study showed SA-β-Gal staining in figure 1A. We added following sentence in Material and Methods section. The same cell line used by Machihara et al. [32] was also employed in the present study. Line 873-874.

Comment: Line 574: It is imperative to specify the number of animals utilized in each experimental group.

Response: The revised text specifies the number of animals used in each experimental group.

Comment: Line 573: It is imperative that the complete environmental conditions for mouse rearing be supplied, including but not limited to temperature and humidity.

Response: We have added the complete environmental conditions for mouse rearing, including temperature and humidity, to the text. Line 987-988.

Comment: Line 583: It should be noted that kidney tissues were also collected during the sampling process.

Response: We appreciate the comment. We have simply added this information into the text. Line 994.

Comment: Line 594: A discrepancy exists between the title and the content: The method description utilizes the term "serum triglycerides" for the measurement of bodily substances, while the title makes reference to "IL-6" without providing any further elaboration or context within the main text. A thorough revision of this text is therefore essential.

Response: Thank you for pointing that out. I've made the correction. Line 1037.

Comment: Line 623: It is imperative that the format of company addresses be standardized. To that end, either detailed addresses must be provided for all companies or only the company names and their respective countries must be included.

Response: The company addresses have been standardized by including only the company names and their respective countries.

Comment: It has been observed that the images contained within the file titled "IDR-3869893-original-images" appear to be of suboptimal clarity.

Response: Thank you for pointing this out. The images in "IDR-3869893-original-images" have been replaced with higher-clarity versions.

Reviewer 2 Report

Comments and Suggestions for Authors

This is a highly compelling and well-structured study that addresses a significant issue in aging populations: intestinal barrier dysfunction. The authors present a novel approach by developing a sustainably produced, ion-exchanged polysaccharide (ulvan-Na) from the fast-growing seaweed Ulva meridionalis. The research is logically sound, moving from sustainable production and compound modification to in vitro anti-aging assays, in vivo validation in a natural aging model, and mechanistic investigations. The data are robust and generally support the conclusions. The potential of ulvan-Na as a functional food ingredient for promoting gut health in the elderly is significant and well-argued.

Comments:

  1. The chosen dose of 50 mg/kg for the mouse study is stated but not justified. A brief explanation (e.g., based on preliminary dose-response experiments or literature on other polysaccharides) would be helpful.
  2. Page 6, Figure 1 Legend, Line 216: "using their RNA" – This phrase is incomplete. It should likely read "The cells were lysed, and their RNA was then subjected to qPCR."

Author Response

Comments and Suggestions for Authors

This is a highly compelling and well-structured study that addresses a significant issue in aging populations: intestinal barrier dysfunction. The authors present a novel approach by developing a sustainably produced, ion-exchanged polysaccharide (ulvan-Na) from the fast-growing seaweed Ulva meridionalis. The research is logically sound, moving from sustainable production and compound modification to in vitro anti-aging assays, in vivo validation in a natural aging model, and mechanistic investigations. The data are robust and generally support the conclusions. The potential of ulvan-Na as a functional food ingredient for promoting gut health in the elderly is significant and well-argued.

I sincerely appreciate your excellent evaluation.

Comments:

  1. The chosen dose of 50 mg/kg for the mouse study is stated but not justified. A brief explanation (e.g., based on preliminary dose-response experiments or literature on other polysaccharides) would be helpful.

Response: Thank you for your comment. The dose of 50 mg/kg was selected based on previous report demonstrating the physiological activity of ulvan at this level. Line 503-505.

  1. Page 6, Figure 1 Legend, Line 216: "using their RNA" – This phrase is incomplete. It should likely read "The cells were lysed, and their RNA was then subjected to qPCR."

Response: Along with your comment, "The cells were lysed, and their RNA was then subjected to qPCR." Line 429.

Reviewer 3 Report

Comments and Suggestions for Authors

Ulvan-Na, an ulvan-polysaccharide subjected to Na+ cation exchange, improves symptoms related to age-related leaky gut.

This is an interesting work of applied biotechnology with major implications for society, such as improving quality of life. However, some comments should be taken into account in order to improve the manuscript.

Lines 36-37, include in global terms which countries have a higher life expectancy than others, comparing this with the Sustainable Development Goals of the 2030 Agenda.

Describe the physiological processes of ulva meridionalis in the introduction, as well as the cation exchange process, as the authors do not go into detail about this.

A flow diagram of the process used should be included in the materials and methods section.

Table 2, primer list, should be included as supplementary material.

Indicate the origin and analytical purity of the reagents.

The results are well structured and discussed, but the figures lack sufficient resolution. The resolution, as well as the size and font of the legend, should be improved.

The conclusions are sound, but the authors should emphasize the scope of the work more in terms of dietary implementation.

The references do not follow MDPI guidelines. The journal title should be italicized, the year should be bold, and the journal's DOI should be included.

Author Response

Ulvan-Na, an ulvan-polysaccharide subjected to Na+ cation exchange, improves symptoms related to age-related leaky gut.

This is an interesting work of applied biotechnology with major implications for society, such as improving quality of life. However, some comments should be taken into account in order to improve the manuscript.

Comment: Lines 36-37, include in global terms which countries have a higher life expectancy than others, comparing this with the Sustainable Development Goals of the 2030 Agenda.

Response: Thank you for the suggestion. The Introduction has been revised to include global differences in life expectancy and their relevance to the Sustainable Development Goals of the 2030 Agenda. Line 38-66.

Comment: Describe the physiological processes of ulva meridionalis in the introduction, as well as the cation exchange process, as the authors do not go into detail about this.

Response: Thank you for the comment. The physiological processes of Ulva meridionalis have been described in the Introduction. To avoid making the Introduction too long, the cation exchange process is briefly mentioned there, with detailed information provided in the Materials and Methods section. Line 216-218, line 220-222.

Comment: A flow diagram of the process used should be included in the materials and methods section.

Response: Thank you for the suggestion. A diagram of the process of ion changed ulvan has been added Figure 8 to the Materials and Methods section in revised manuscript.

Comment: Table 2, primer list, should be included as supplementary material.

Response: Thank you for the suggestion. Table 2 (primer list) has been moved to the supplementary material as supplementary table 1.

Comment: Indicate the origin and analytical purity of the reagents.

Response: Thank you for the comment. The analytical purity of ulvan have been indicated in the Methods section. Line 843-846.

Comment: The results are well structured and discussed, but the figures lack sufficient resolution. The resolution, as well as the size and font of the legend, should be improved.

Response: Thank you for the comment. The resolution, size and font size of all figures have been improved.

Comment: The conclusions are sound, but the authors should emphasize the scope of the work more in terms of dietary implementation.

Response: The conclusion has been revised to emphasize dietary implementation while noting that the findings are limited to animal experiments. Line 871-874.

Comment: The references do not follow MDPI guidelines. The journal title should be italicized, the year should be bold, and the journal's DOI should be included.

Response: Thank you for the comment. The references have been revised to follow Marine drugs guidelines with journal titles italicized and years in bold.

Reviewer 4 Report

Comments and Suggestions for Authors

Ulvan-Na, an ulvan-polysaccharide subjected to Na+ cation exchange, improves symptoms related to age-related leaky gut.

Great work, solid paper; there are some organizational issues.

Revise the title

Revise the abstract: Add experimental design, statement of statistical analysis, and P values for significant terms. Also, you need to add information about the animals, such as age, number, treatments, etc.

Revise the introduction: It's too long; just focus on the topics related to the work.

Be consistent with abbreviations and italicize them in the whole document, U. meridionalis, U. prolifera, etc.

Try to avoid "we" and "our"

Materials and methods: Try to shorten; too many unnecessary details. Avoid too many "we and our".

The result section: Reorganize, some materials in here belong to the M&M section, others are related to the discussion.

Author Response

Ulvan-Na, an ulvan-polysaccharide subjected to Na+ cation exchange, improves symptoms related to age-related leaky gut.

Great work, solid paper; there are some organizational issues.

Comment: Revise the title

Response: We revised the title as follows. Ulvan-Na, an ulvan subjected to Na+ cation exchange, improves intestinal barrier function in age-related leaky gut.

Comment: Revise the abstract: Add experimental design, statement of statistical analysis, and P values for significant terms. Also, you need to add information about the animals, such as age, number, treatments, etc.

Response: Thank you for the comment. The abstract has been revised to include the experimental design, statistical analysis, P values, and details of the animals (age, number, and treatments).

Comment: Revise the introduction: It's too long; just focus on the topics related to the work.

Response: Thank you for the comment. The Introduction has been shortened to focus only on topics directly related to this work.

Comment: Be consistent with abbreviations and italicize them in the whole document, U. meridionalis, U. prolifera, etc.

Response: Thank you for pointing this out. The abbreviations have been made consistent, and species names such as U. meridionalis and U. prolifera have been italicized throughout the revised manuscript.

 Comment: Try to avoid "we" and "our"

Response: Thank you for the suggestion. All instances of “we” and “our” have been revised to maintain an objective style.

Comment: Materials and methods: Try to shorten; too many unnecessary details. Avoid too many "we and our".

Response: The Materials and Methods section has been shortened by removing unnecessary details, and the use of “we” and “our” has been minimized to maintain an objective tone.

Comment: The result section: Reorganize, some materials in here belong to the M&M section, others are related to the discussion.

Response: The experimental methods and discussion that were previously included in the Results section have been moved to the appropriate sections. In particular, the cultivation methods described in Section 2.1 have been added to the Materials and Methods section, while the discussion in Section 2.9 was removed because it was already described in the Discussion section and was therefore redundant.

Reviewer 5 Report

Comments and Suggestions for Authors

The manuscript titled "Ulvan-Na, an ulvan polysaccharide subjected to Na+ cation exchange, improves symptoms related to age-related leaky gut." addresses a relevant topic and is appropriate for this journal.

The manuscript is well-written, well-structured, and well-founded.

I indicate below any necessary corrections, especially regarding the taxonomy/nomenclature rules of the cited algae.
Authors should always write genus and species names in italics.

Corrections needed:

line 102 - Ulva meridionalis, a green alga of the genus Ulva, is a very fast-growing seaweed (Note: "Ulva meridionalis" and "Ulva" are in italics)

line 116 - In this study, we focused on U. meridionalis as a sustainable material and investi- (Note: "U. meridionalis" is in italics)

line 117 - gated land-based open-air aquaculture methods to achieve stable production of U. merid- (Note: "U. merid" is in talics)

line 118 - ionalis. We subsequently aimed to develop a health-promoting material derived from U. (Note: "ionalis" and "U." are in italics)

line 119 - meridionalis polysaccharides to improve age-related leaky gut symptoms and progres- (Note: "meridionalis "is in italics)

line 123 - U. meridionalis, similar in morphology to Ulva prolifera in terms of its thin and (Note: "U. meridionalis" and "Ulva prolifera" rae in italics)

line 125 - cell morphology and DNA marker comparisons [31]. Since the mass of U. meridionalis (Note: "U. meridionalis" is in italics)

line 128 - Therefore, we investigated land-based cultivation methods for "U. meridionalis". Our re- (Note: "U. meridionalis" is in italics)

line 130 - niques for the green alga "U. prolifera". With these methods, we investigated land-based (Note: "U. meridionalis" is in italics)

line 131 - cultivation for U. meridionalis. For seedling production, we used the "germling cluster" (Note: "U. meridionalis" is in italics)

line 137 - provided to ensure adequate mixing of "U. meridionalis". The system was configured to (Note: "U. meridionalis" is in italics)

line 139 - S1A). Figure S1B shows the growth curves of "U. meridionalis" in the 1-ton tanks over two (Note: "U. meridionalis" is in italics)

line 143 - the harvested U. meridionalis is shown in Figure S1C. These results demonstrate that mass (Note: "U. meridionalis" is in italics)

line 144 - cultivation of U. meridionalis using the germling cluster method is feasible. (Note: "U. meridionalis" is in italics)

line 146 - Previous reports have indicated that U. meridionalis fixes COâ‚‚-derived carbon in pol- (Note: "U. meridionalis" is in italics)

line 148 - tribute to carbon dioxide mitigation efforts. The harvested U. meridionalis (Figure S1C) (Note: "U. meridionalis" is in italics)

line 153 - van" refers to the water-soluble sulfated polysaccharides produced by different Ulva spe- (Note: "Ulva" is in italics)

line 156 - that the monosaccharide composition of ulvan from the U. meridionalis strain used in this (Note: "U. meridionalis" is in italics)

line 161 - of ulvan derived from U. meridionalis remain unexplored. Considering the potential anti- (Note: "U. meridionalis" is in italics)

line 244 -  2D). In particular, aged mice presented significant reductions in the abundances of Alisti- (Note: "Alisti" is in italics)

line 245 -  pes, Parabacteroides, Clostridiales vadin BB60, Lachnoclostridium, and ASF356 (pes", "Parabacteroides" and "Lachnoclostridium" are in italics)

line 413 -  In this study, we established an outdoor mass aquaculture method for U. merid- (Note: "U. merid" is in italics)

line 414 -  ionalis to produce ulvan-Na by modifying the cation content of the water-soluble poly- (Note: "ionalis" is in italics)

line 422 - First, we demonstrated that mass aquaculture of U. meridionalis is possible using the (Note: "U. meridionalis" is in italics)

line 425 - the most efficient land-based aquaculture technique [26]. In our previous studies, U. me- (Note: "me" is in italics)

line 426 - ridionalis exhibited a daily growth rate of approximately fourfold under controlled labor- (Note: "ridionalis" is in italics)

line 430 - U. prolifera, which responds to an established outdoor cultivation method with a daily (Note: "U. prolifera" is in italics)

line 434 - average daily growth rate of approximately 1.1-fold [46]. Therefore, U. meridionalis grows (Note: "U. meridionalis" is in italics)

line 438 - rived from U. meridionalis. Conventional ulvan did not have antiaging effects; however, (Note: "U. meridionalis" is in italics)

line 512 - The well-washed U. meridionalis macroalga was dried and powdered. The powder (Note: "U. meridionalis" is in italics)

line 534 - MEMα medium (Wako, Tokyo, Japan) supplemented with 10% FBS, 100 U/mL penicillin,

line 535 - and 100 µg/mL streptomycin. The cell cultures were passaged at a 1:4 ratio every three

line 644 - This study demonstrated that mass aquaculture of U. meridionalis through land- (Note: "U. meridionalis" is in italics)

Author Response

The manuscript titled "Ulvan-Na, an ulvan polysaccharide subjected to Na+ cation exchange, improves symptoms related to age-related leaky gut." addresses a relevant topic and is appropriate for this journal.

The manuscript is well-written, well-structured, and well-founded.

I indicate below any necessary corrections, especially regarding the taxonomy/nomenclature rules of the cited algae.
Authors should always write genus and species names in italics.Corrections needed:

Response: Thank you for your comments. I have addressed all the points you brought up.

Comment: line 102 - Ulva meridionalis, a green alga of the genus Ulva, is a very fast-growing seaweed (Note: "Ulva meridionalis" and "Ulva" are in italics)

line 116 - In this study, we focused on U. meridionalis as a sustainable material and investi- (Note: "U. meridionalis" is in italics)

line 117 - gated land-based open-air aquaculture methods to achieve stable production of U. merid- (Note: "U. merid" is in talics)

line 118 - ionalis. We subsequently aimed to develop a health-promoting material derived from U. (Note: "ionalis" and "U." are in italics)

line 119 - meridionalis polysaccharides to improve age-related leaky gut symptoms and progres- (Note: "meridionalis "is in italics)

line 123 - U. meridionalis, similar in morphology to Ulva prolifera in terms of its thin and (Note: "U. meridionalis" and "Ulva prolifera" rae in italics)

line 125 - cell morphology and DNA marker comparisons [31]. Since the mass of U. meridionalis (Note: "U. meridionalis" is in italics)

line 128 - Therefore, we investigated land-based cultivation methods for "U. meridionalis". Our re- (Note: "U. meridionalis" is in italics)

line 130 - niques for the green alga "U. prolifera". With these methods, we investigated land-based (Note: "U. meridionalis" is in italics)

line 131 - cultivation for U. meridionalis. For seedling production, we used the "germling cluster" (Note: "U. meridionalis" is in italics)

line 137 - provided to ensure adequate mixing of "U. meridionalis". The system was configured to (Note: "U. meridionalis" is in italics)

line 139 - S1A). Figure S1B shows the growth curves of "U. meridionalis" in the 1-ton tanks over two (Note: "U. meridionalis" is in italics)

line 143 - the harvested U. meridionalis is shown in Figure S1C. These results demonstrate that mass (Note: "U. meridionalis" is in italics)

line 144 - cultivation of U. meridionalis using the germling cluster method is feasible. (Note: "U. meridionalis" is in italics)

line 146 - Previous reports have indicated that U. meridionalis fixes COâ‚‚-derived carbon in pol- (Note: "U. meridionalis" is in italics)

line 148 - tribute to carbon dioxide mitigation efforts. The harvested U. meridionalis (Figure S1C) (Note: "U. meridionalis" is in italics)

line 153 - van" refers to the water-soluble sulfated polysaccharides produced by different Ulva spe- (Note: "Ulva" is in italics)

line 156 - that the monosaccharide composition of ulvan from the U. meridionalis strain used in this (Note: "U. meridionalis" is in italics)

line 161 - of ulvan derived from U. meridionalis remain unexplored. Considering the potential anti- (Note: "U. meridionalis" is in italics)

line 244 -  2D). In particular, aged mice presented significant reductions in the abundances of Alisti- (Note: "Alisti" is in italics)

line 245 -  pes, Parabacteroides, Clostridiales vadin BB60, Lachnoclostridium, and ASF356 (pes", "Parabacteroides" and "Lachnoclostridium" are in italics)

line 413 -  In this study, we established an outdoor mass aquaculture method for U. merid- (Note: "U. merid" is in italics)

line 414 -  ionalis to produce ulvan-Na by modifying the cation content of the water-soluble poly- (Note: "ionalis" is in italics)

line 422 - First, we demonstrated that mass aquaculture of U. meridionalis is possible using the (Note: "U. meridionalis" is in italics)

line 425 - the most efficient land-based aquaculture technique [26]. In our previous studies, U. me- (Note: "me" is in italics)

line 426 - ridionalis exhibited a daily growth rate of approximately fourfold under controlled labor- (Note: "ridionalis" is in italics)

line 430 - U. prolifera, which responds to an established outdoor cultivation method with a daily (Note: "U. prolifera" is in italics)

line 434 - average daily growth rate of approximately 1.1-fold [46]. Therefore, U. meridionalis grows (Note: "U. meridionalis" is in italics)

line 438 - rived from U. meridionalis. Conventional ulvan did not have antiaging effects; however, (Note: "U. meridionalis" is in italics)

line 512 - The well-washed U. meridionalis macroalga was dried and powdered. The powder (Note: "U. meridionalis" is in italics)

line 534 - MEMα medium (Wako, Tokyo, Japan) supplemented with 10% FBS, 100 U/mL penicillin,

line 535 - and 100 µg/mL streptomycin. The cell cultures were passaged at a 1:4 ratio every three

line 644 - This study demonstrated that mass aquaculture of U. meridionalis through land- (Note: "U. meridionalis" is in italics)

Round 2

Reviewer 1 Report

Comments and Suggestions for Authors

It is evident that the logical structure of the paper is lacking in clarity and cohesion. It is evident that the link between the two animal experiments has not been elucidated with sufficient clarity. The rationale behind the selection of distinct ages for the two experimental mice remains unaddressed.

It is imperative that more significant results are articulated in the abstract, as opposed to the inclusion of experimental methods. For instance, which specific microbial communities were regulated, and which inflammatory factors were affected?

Figure S3 is not clear.

Author Response

Comment: It is evident that the logical structure of the paper is lacking in clarity and cohesion. It is evident that the link between the two animal experiments has not been elucidated with sufficient clarity. The rationale behind the selection of distinct ages for the two experimental mice remains unaddressed.

Response: Thank you for your comment. Although mice of different ages were used, both groups exhibited features characteristic of age-dependent leaky gut. First, we used older mice to confirm the presence of leaky gut and identify reliable indicators. Having verified that the same indicators were altered in 74-week-old mice, we selected this age for evaluating ulvan-Na, as it enabled the model to be established within a shorter experimental timeframe. Line 753-777.

Comment: It is imperative that more significant results are articulated in the abstract, as opposed to the inclusion of experimental methods. For instance, which specific microbial communities were regulated, and which inflammatory factors were affected?

Response: Thank you for your insightful comment. In the revised manuscript, we have added more significant results to the abstract. With respect to the regulation of specific microbial communities and inflammatory factors, direct verification was not performed in this study; therefore, we have addressed this limitation and added a corresponding statement in the Discussion section “In this study, we verified that ulvan-Na administration altered the composition of the gut microbiota. However, we could not determine the direct impact of specific microbial groups on the intestinal environment. This limitation is an important topic for future research”. Line 29-32, and line 808-811.

Comment: Figure S3 is not clear.

Response: We have improved the resolution and clarity of Figure S3 in the revised version.

Round 3

Reviewer 1 Report

Comments and Suggestions for Authors

'To evaluateits effects, aged C57BL/6 mice (74-week-old) were orally administered D.W. (n=5), ulvan(n=4) or ulvan-Na (n=4) daily for 25 days. Statistical significance was determined usingTukey's test, with P<0.05 considered significant. ' should be deleted.

'(P < 0.05)' in abstract should be deleted.

The results in the abstract should be presented in the order of the results in the main text.

The lines 753-777 and 808-811 mentioned in the author's previous response do not exist in the main text.

Author Response

Comment: 'To evaluateits effects, aged C57BL/6 mice (74-week-old) were orally administered D.W. (n=5), ulvan(n=4) or ulvan-Na (n=4) daily for 25 days. Statistical significance was determined usingTukey's test, with P<0.05 considered significant. ' should be deleted.

Response: Thank you for your suggestion. We have deleted the indicated sentence in the revised manuscript.

Comment: '(P < 0.05)' in abstract should be deleted.

Response: We have deleted the indicated word in the revised manuscript.

Comment: The results in the abstract should be presented in the order of the results in the main text.

Response: Thank you for your comment. We have revised the abstract so that the results are presented in the same order as in the main text.

Comment: The lines 753-777 and 808-811 mentioned in the author's previous response do not exist in the main text.

Response: Thank you for pointing this out, and we sincerely apologize for the confusion. We inadvertently cited incorrect line numbers in our previous response. The correct locations are lines 760–784 (instead of 753–777) and lines 805–809 (instead of 808–811) in the revised manuscript.

lines 760–784 : “This study confirmed colonic tissue damage in aged mice and demonstrated weakened intestinal barrier function through assessment of dextran-FITC translocation from the intestine into the bloodstream, a common indicator of barrier integrity. In this study, mice aged 74 to 100 weeks were used as an aged mouse model. All mice exhibited elevated blood LPS concentrations and increased intestinal permeability of 4 kDa FITC-dextran into the bloodstream, confirming the presence of leaky gut. Therefore, these mice were used as an age-dependent leaky gut model.”

lines 805–809: “In this study, we verified that ulvan-Na administration altered the composition of the gut microbiota. However, we could not determine the direct impact of specific microbial groups on the intestinal environment. This limitation is an important topic for future research.